# Orthogonal Random Features

**Felix Xinnan Yu   Ananda Theertha Suresh   Krzysztof Choromanski**
**Daniel Holtmann-Rice   Sanjiv Kumar**

Google Research, New York
{felixyu, theertha, kchoro, dhr, sanjivk}@google.com

## Abstract

We present an intriguing discovery related to Random Fourier Features: in Gaussian kernel approximation, replacing the random Gaussian matrix by a properly scaled random orthogonal matrix significantly decreases kernel approximation error. We call this technique Orthogonal Random Features (ORF), and provide theoretical and empirical justification for this behavior. Motivated by this discovery, we further propose Structured Orthogonal Random Features (SORF), which uses a class of structured discrete orthogonal matrices to speed up the computation. The method reduces the time cost from $\mathcal{O}(d^2)$ to $\mathcal{O}(d \log d)$, where $d$ is the data dimensionality, with almost no compromise in kernel approximation quality compared to ORF. Experiments on several datasets verify the effectiveness of ORF and SORF over the existing methods. We also provide discussions on using the same type of discrete orthogonal structure for a broader range of applications.

## 1   Introduction

Kernel methods are widely used in nonlinear learning [8], but they are computationally expensive for large datasets. Kernel approximation is a powerful technique to make kernel methods scalable, by mapping input features into a new space where dot products approximate the kernel well [19]. With accurate kernel approximation, efficient linear classifiers can be trained in the transformed space while retaining the expressive power of nonlinear methods [10, 21].

Formally, given a kernel $K(\cdot, \cdot) : \mathbb{R}^d \times \mathbb{R}^d \to \mathbb{R}$, kernel approximation methods seek to find a nonlinear transformation $\phi(\cdot) : \mathbb{R}^d \to \mathbb{R}^{d'}$ such that, for any $\mathbf{x}, \mathbf{y} \in \mathbb{R}^d$

$$K(\mathbf{x}, \mathbf{y}) \approx \hat{K}(\mathbf{x}, \mathbf{y}) = \phi(\mathbf{x})^T \phi(\mathbf{y}).$$

Random Fourier Features [19] are used widely in approximating smooth, shift-invariant kernels. This technique requires the kernel to exhibit two properties: 1) shift-invariance, *i.e.* $K(\mathbf{x}, \mathbf{y}) = K(\Delta)$ where $\Delta = \mathbf{x} - \mathbf{y}$; and 2) positive semi-definiteness of $K(\Delta)$ on $\mathbb{R}^d$. The second property guarantees that the Fourier transform of $K(\Delta)$ is a nonnegative function [3]. Let $p(\mathbf{w})$ be the Fourier transform of $K(\mathbf{z})$. Then,

$$K(\mathbf{x} - \mathbf{y}) = \int_{\mathbb{R}^d} p(\mathbf{w}) e^{j \mathbf{w}^T (\mathbf{x} - \mathbf{y})} d\mathbf{w}.$$

This means that one can treat $p(\mathbf{w})$ as a density function and use Monte-Carlo sampling to derive the following nonlinear map for a real-valued kernel:

$$\phi(\mathbf{x}) = \sqrt{1/D} \big[ \sin(\mathbf{w}_1^T \mathbf{x}), \cdots, \sin(\mathbf{w}_D^T \mathbf{x}), \cos(\mathbf{w}_1^T \mathbf{x}), \cdots, \cos(\mathbf{w}_D^T \mathbf{x}) \big]^T,$$

where $\mathbf{w}_i$ is sampled i.i.d. from a probability distribution with density $p(\mathbf{w})$. Let $\mathbf{W} = \big[ \mathbf{w}_1, \cdots, \mathbf{w}_D \big]^T$. The linear transformation $\mathbf{W}\mathbf{x}$ is central to the above computation since,

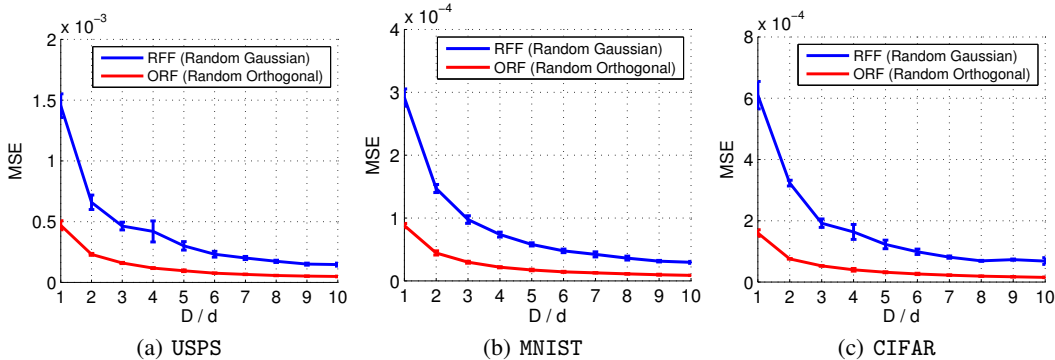

Figure 1: Kernel approximation mean squared error (MSE) for the Gaussian kernel $K(\mathbf{x}, \mathbf{y}) = e^{-||\mathbf{x}-\mathbf{y}||^2/2\sigma^2}$. $D$: number of rows in the linear transformation $\mathbf{W}$. $d$: input dimension. ORF imposes orthogonality on $\mathbf{W}$ (Section 3).

- The choice of matrix $\mathbf{W}$ determines how well the estimated kernel converges to the actual kernel;
- The computation of $\mathbf{Wx}$ has space and time costs of $\mathcal{O}(Dd)$. This is expensive for high-dimensional data, especially since $D$ is often required to be larger than $d$ to achieve low approximation error.

In this work, we address both of the above issues. We first show an intriguing discovery (Figure 1): by enforcing orthogonality on the rows of $\mathbf{W}$, the kernel approximation error can be significantly reduced. We call this method Orthogonal Random Features (ORF). Section 3 describes the method and provides theoretical explanation for the improved performance.

Since both generating a $d \times d$ orthogonal matrix ($\mathcal{O}(d^3)$ time and $\mathcal{O}(d^2)$ space) and computing the transformation ($\mathcal{O}(d^2)$ time and space) are prohibitively expensive for high-dimensional data, we further propose Structured Orthogonal Random Features (SORF) in Section 4. The idea is to replace random orthogonal matrices by a class of special structured matrices consisting of products of binary diagonal matrices and Walsh-Hadamard matrices. SORF has fast computation time, $\mathcal{O}(D \log d)$, and almost no extra memory cost (with efficient in-place implementation). We show extensive experiments in Section 5. We also provide theoretical discussions in Section 6 of applying the structured matrices in a broader range of applications where random Gaussian matrix is used.

## 2 Related Works

Explicit nonlinear random feature maps have been constructed for many types of kernels, such as intersection kernels [15], generalized RBF kernels [22], skewed multiplicative histogram kernels [14], additive kernels [24], and polynomial kernels [11, 18]. In this paper, we focus on approximating Gaussian kernels following the seminal Random Fourier Features (RFF) framework [19], which has been extensively studied both theoretically and empirically [26, 20, 23].

Key to the RFF technique is Monte-Carlo sampling. It is well known that the convergence of Monte-Carlo can be largely improved by carefully choosing a deterministic sequence instead of random samples [17]. Following this line of reasoning, Yang et al. [25] proposed to use low-displacement rank sequences in RFF. Yu et al. [28] studied optimizing the sequences in a data-dependent fashion to achieve more compact maps. In contrast to the above works, this paper is motivated by an intriguing new discovery that using orthogonal random samples provides much faster convergence. Compared to [25], the proposed SORF method achieves both lower kernel approximation error and greatly reduced computation and memory costs. Furthermore, unlike [28], the results in this paper are data independent.

Structured matrices have been used for speeding up dimensionality reduction [1], binary embedding [27], deep neural networks [5] and kernel approximation [13, 28, 7]. For the kernel approximation works, in particular, the "structured randomness" leads to a minor loss of accuracy, but allows faster computation since the structured matrices enable the use of FFT-like algorithms. Furthermore, these matrices provide substantial model compression since they require subquadratic (usually only linear)

| Method | Extra Memory | Time | Lower error than RFF? |
|---|---|---|---|
| Random Fourier Feature (RFF) [19] | $\mathcal{O}(Dd)$ | $\mathcal{O}(Dd)$ | - |
| Compact Nonlinear Map (CNM) [28] | $\mathcal{O}(Dd)$ | $\mathcal{O}(Dd)$ | Yes (data-dependent) |
| Quasi-Monte Carlo (QMC) [25] | $\mathcal{O}(Dd)$ | $\mathcal{O}(Dd)$ | Yes |
| Structured (fastfood/circulant) [28, 13] | $\mathcal{O}(D)$ | $\mathcal{O}(D \log d)$ | No |
| **Orthogonal Random Feature (ORF)** | $\mathcal{O}(Dd)$ | $\mathcal{O}(Dd)$ | **Yes** |
| **Structured ORF (SORF)** | $\mathcal{O}(D)$ or $\mathcal{O}(1)$ | $\mathcal{O}(D \log d)$ | **Yes** |

Table 1: Comparison of different kernel approximation methods under the framework of Random Fourier Features [19]. We assume $D \geq d$. The proposed SORF method have $\mathcal{O}(D)$ degrees of freedom. The computations can be efficiently implemented as in-place operations with fixed random seeds. Therefore it can cost $\mathcal{O}(1)$ in extra space.

space. In comparison with the above works, our proposed methods SORF and ORF are more effective than RFF. In particular SORF demonstrates *both* lower approximation error and better efficiency than RFF. Table 1 compares the space and time costs of different techniques.

## 3 Orthogonal Random Features

Our goal is to approximate a Gaussian kernel of the form

$$K(\mathbf{x}, \mathbf{y}) = e^{-||\mathbf{x}-\mathbf{y}||^2/2\sigma^2}.$$

In the paragraph below, we assume a square linear transformation matrix $\mathbf{W} \in \mathbb{R}^{D \times d}$, $D = d$. When $D < d$, we simply use the first $D$ dimensions of the result. When $D > d$, we use multiple independently generated random features and concatenate the results. We comment on this setting at the end of this section.

Recall that the linear transformation matrix of RFF can be written as

$$\mathbf{W}_{\text{RFF}} = \frac{1}{\sigma}\mathbf{G}, \tag{1}$$

where $\mathbf{G} \in \mathbb{R}^{d \times d}$ is a random Gaussian matrix, with every entry sampled independently from the standard normal distribution. Denote the approximate kernel based on the above $\mathbf{W}_{\text{RFF}}$ as $K_{\text{RFF}}(\mathbf{x}, \mathbf{y})$. For completeness, we first show the expectation and variance of $K_{\text{RFF}}(\mathbf{x}, \mathbf{y})$.

**Lemma 1.** *(Appendix A.2)* $K_{RFF}(\mathbf{x}, \mathbf{y})$ *is an unbiased estimator of the Gaussian kernel, i.e.,* $\mathbb{E}(K_{RFF}(\mathbf{x}, \mathbf{y})) = e^{-||\mathbf{x}-\mathbf{y}||^2/2\sigma^2}$. *Let* $z = ||\mathbf{x} - \mathbf{y}||/\sigma$. *The variance of* $K_{RFF}(\mathbf{x}, \mathbf{y})$ *is* $Var(K_{RFF}(\mathbf{x}, \mathbf{y})) = \frac{1}{2D}\left(1 - e^{-z^2}\right)^2$.

The idea of Orthogonal Random Features (ORF) is to impose orthogonality on the matrix on the linear transformation matrix $\mathbf{G}$. Note that one cannot achieve unbiased kernel estimation by simply replacing $\mathbf{G}$ by an orthogonal matrix, since the norms of the rows of $\mathbf{G}$ follow the $\chi$-distribution, while rows of an orthogonal matrix have the unit norm. The linear transformation matrix of ORF has the following form

$$\mathbf{W}_{\text{ORF}} = \frac{1}{\sigma}\mathbf{S}\mathbf{Q}, \tag{2}$$

where $\mathbf{Q}$ is a uniformly distributed random orthogonal matrix[1]. The set of rows of $\mathbf{Q}$ forms a bases in $\mathbb{R}^d$. $\mathbf{S}$ is a diagonal matrix, with diagonal entries sampled i.i.d. from the $\chi$-distribution with $d$ degrees of freedom. $\mathbf{S}$ makes the norms of the rows of $\mathbf{S}\mathbf{Q}$ and $\mathbf{G}$ identically distributed.

Denote the approximate kernel based on the above $\mathbf{W}_{\text{ORF}}$ as $K_{\text{ORF}}(\mathbf{x}, \mathbf{y})$. The following shows that $K_{\text{ORF}}(\mathbf{x}, \mathbf{y})$ is an unbiased estimator of the kernel, and it has lower variance in comparison to RFF.

**Theorem 1.** $K_{ORF}(\mathbf{x}, \mathbf{y})$ *is an unbiased estimator of the Gaussian kernel, i.e.,*

$$\mathbb{E}(K_{ORF}(\mathbf{x}, \mathbf{y})) = e^{-||\mathbf{x}-\mathbf{y}||^2/2\sigma^2}.$$

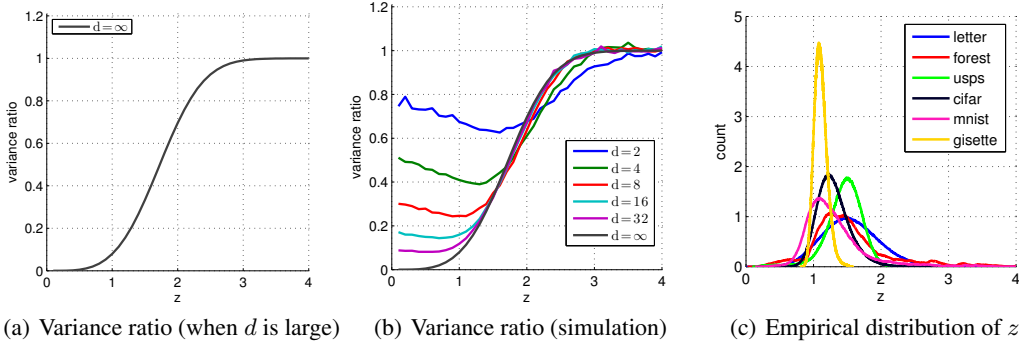

(a) Variance ratio (when $d$ is large)   (b) Variance ratio (simulation)   (c) Empirical distribution of $z$

Figure 2: (a) $\mathrm{Var}(K_{\mathrm{ORF}}(\mathbf{x}, \mathbf{y}))/\mathrm{Var}(K_{\mathrm{RFF}}(\mathbf{x}, \mathbf{y}))$ when $d$ is large and $d = D$. $z = ||\mathbf{x} - \mathbf{y}||/\sigma$. (b) Simulation of $\mathrm{Var}(K_{\mathrm{ORF}}(\mathbf{x}, \mathbf{y}))/\mathrm{Var}(K_{\mathrm{RFF}}(\mathbf{x}, \mathbf{y}))$ when $D = d$. Note that the empirical variance is the Mean Squared Error (MSE). (c) Distribution of $z$ for several datasets, when we set $\sigma$ as the mean distance to 50th-nearest neighbor for samples from the dataset. The count is normalized such that the area under curve for each dataset is 1. Observe that most points in all the datasets have $z < 2$. As shown in (a), for these values of $z$, ORF has much smaller variance compared to the standard RFF.

*Let $D \leq d$, and $z = ||\mathbf{x} - \mathbf{y}||/\sigma$. There exists a function $f$ such that for all $z$, the variance of $K_{ORF}(\mathbf{x}, \mathbf{y})$ is bounded by*

$$Var\left(K_{ORF}(\mathbf{x}, \mathbf{y})\right) \leq \frac{1}{2D}\left(\left(1 - e^{-z^2}\right)^2 - \frac{D-1}{d}e^{-z^2}z^4\right) + \frac{f(z)}{d^2}.$$

*Proof.* We first show the proof of the unbiasedness. Let $\mathbf{z} = \frac{\mathbf{x} - \mathbf{y}}{\sigma}$, and $z = ||\mathbf{z}||$, then $\mathbb{E}(K_{ORF}(\mathbf{x}, \mathbf{y})) = \mathbb{E}\left(\frac{1}{D}\sum_{i=1}^{D}\cos(\mathbf{w}_i^T \mathbf{z})\right) = \frac{1}{D}\sum_{i=1}^{D}\mathbb{E}\left(\cos(\mathbf{w}_i^T \mathbf{z})\right)$. Based on the definition of ORF, $\mathbf{w}_1, \mathbf{w}_2, \ldots, \mathbf{w}_D$ are $D$ random vectors given by $\mathbf{w}_i = s_i \mathbf{u}_i$, with $\mathbf{u}_1, \mathbf{u}_2, \ldots, \mathbf{u}_d$ a uniformly chosen random orthonormal basis for $\mathbf{R}^d$, and $s_i$'s are independent $\chi$-distributed random variables with $d$ degrees of freedom. It is easy to show that for each $i$, $\mathbf{w}_i$ is distributed according to $N(0, \mathbf{I}_d)$, and hence by Bochner's theorem,

$$\mathbb{E}[\cos(\mathbf{w}^T \mathbf{z})] = e^{-z^2/2}.$$

We now show a proof sketch of the variance. Suppose, $a_i = \cos(\mathbf{w}_i^T \mathbf{z})$.

$$\mathrm{Var}\left(\frac{1}{D}\sum_{i=1}^{D} a_i\right) = \mathbb{E}\left[\left(\frac{\sum_{i=1}^{D} a_i}{D}\right)^2\right] - \mathbb{E}\left[\left(\frac{\sum_{i=1}^{D} a_i}{D}\right)\right]^2$$

$$= \frac{1}{D^2}\sum_i\left(\mathbb{E}[a_i^2] - \mathbb{E}[a_i]^2\right) + \frac{1}{D^2}\sum_i\sum_{j\neq i}\left(\mathbb{E}[a_i a_j] - \mathbb{E}[a_i]\mathbb{E}[a_j]\right)$$

$$= \frac{\left(1 - e^{-z^2}\right)^2}{2D} + \frac{D(D-1)}{D^2}\left(\mathbb{E}[a_1 a_2] - e^{-z^2}\right),$$

where the last equality follows from symmetry. The first term in the resulting expression is exactly the variance of RFF. In order to have lower variance, $\mathbb{E}[a_1 a_2] - e^{-z^2}$ must be negative. We use the following lemma to quantify this term.

**Lemma 2.** *(Appendix A.3) There is a function $f$ such that for any $z$,*

$$\mathbb{E}[a_i a_j] \leq e^{-z^2} - e^{-z^2}\frac{z^4}{2d} + \frac{f(z)}{d^2}. \qquad \square$$

Therefore, for a large $d$, and $D \leq d$, the ratio of the variance of ORF and RFF is

$$\frac{\mathrm{Var}(K_{\mathrm{ORF}}(\mathbf{x}, \mathbf{y}))}{\mathrm{Var}(K_{\mathrm{RFF}}(\mathbf{x}, \mathbf{y}))} \approx 1 - \frac{(D-1)e^{-z^2}z^4}{d(1 - e^{-z^2})^2}. \qquad (3)$$

Figure 2(a) shows the ratio of the variance of ORF to that of RFF when $D = d$ and $d$ is large. First notice that this ratio is always smaller than 1, and hence ORF always provides improvement over

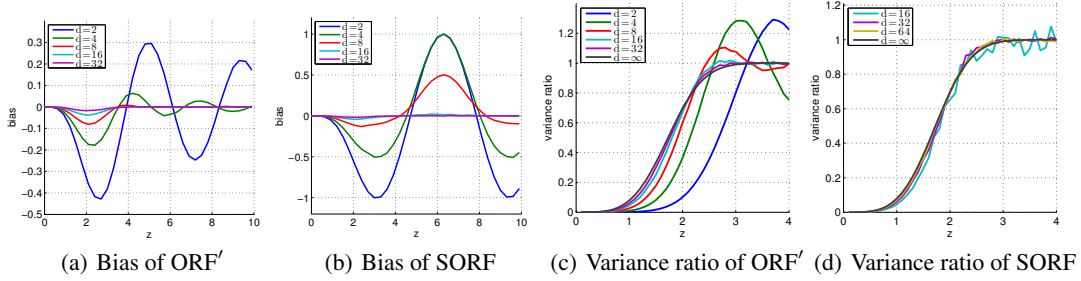

(a) Bias of ORF′     (b) Bias of SORF     (c) Variance ratio of ORF′   (d) Variance ratio of SORF

Figure 3: Simulations of bias and variance of ORF′and SORF. $z = ||\mathbf{x} - \mathbf{y}||/\sigma$. (a) $\mathbb{E}(K_{\mathrm{ORF'}}(\mathbf{x}, \mathbf{y})) - e^{-z^2/2}$. (b) $\mathbb{E}(K_{\mathrm{SORF}}(\mathbf{x}, \mathbf{y})) - e^{-z^2/2}$. (c) $\mathrm{Var}(K_{\mathrm{ORF'}}(\mathbf{x}, \mathbf{y}))/\mathrm{Var}(K_{\mathrm{RFF}}(\mathbf{x}, \mathbf{y}))$. (d) $\mathrm{Var}(K_{\mathrm{SORF}}(\mathbf{x}, \mathbf{y}))/\mathrm{Var}(K_{\mathrm{RFF}}(\mathbf{x}, \mathbf{y}))$. Each point on the curve is based on 20,000 choices of the random matrices and two fixed points with distance $z$. For both ORF and ORF′, even at $d = 32$, the bias is close to 0 and the variance is close to that of $d = \infty$ (Figure 2(a)).

the conventional RFF. Interestingly, we gain significantly for small values of $z$. In fact, when $z \to 0$ and $d \to \infty$, the ratio is roughly $z^2$ (note $e^x \approx 1 + x$ when $x \to 0$), and ORF exhibits infinitely lower error relative to RFF. Figure 2(b) shows empirical simulations of this ratio. We can see that the variance ratio is close to that of $d = \infty$ (3), even when $d = 32$, a fairly low-dimensional setting in real-world cases.

Recall that $z = ||\mathbf{x} - \mathbf{y}||/\sigma$. This means that ORF preserves the kernel value especially well for data points that are close, thereby retaining the local structure of the dataset. Furthermore, empirically $\sigma$ is typically not set too small in order to prevent overfitting—a common rule of thumb is to set $\sigma$ to be the average distance of 50th-nearest neighbors in a dataset. In Figure 2(c), we plot the distribution of $z$ for several datasets with this choice of $\sigma$. These distributions are all concentrated in the regime where ORF yields substantial variance reduction.

The above analysis is under the assumption that $D \leq d$. Empirically, for RFF, $D$ needs to be larger than $d$ in order to achieve low approximation error. In that case, we independently generate and apply the transformation (2) multiple times. The next lemma bounds the variance for this case.

**Corollary 1.** *Let $D = m \cdot d$, for an integer $m$ and $z = ||\mathbf{x} - \mathbf{y}||/\sigma$. There exists a function $f$ such that for all $z$, the variance of $K_{ORF}(\mathbf{x}, \mathbf{y})$ is bounded by*

$$Var\left(K_{ORF}(\mathbf{x}, \mathbf{y})\right) \leq \frac{1}{2D}\left(\left(1 - e^{-z^2}\right)^2 - \frac{d-1}{d}e^{-z^2}z^4\right) + \frac{f(z)}{dD}.$$

## 4 Structured Orthogonal Random Features

In the previous section, we presented Orthogonal Random Features (ORF) and provided a theoretical explanation for their effectiveness. Since generating orthogonal matrices in high dimensions can be expensive, here we propose a fast version of ORF by imposing structure on the orthogonal matrices. This method can provide drastic memory and time savings with minimal compromise on kernel approximation quality. Note that the previous works on fast kernel approximation using structured matrices do not use structured *orthogonal* matrices [13, 28, 7].

Let us first introduce a simplified version of ORF: replace $\mathbf{S}$ in (2) by a scalar $\sqrt{d}$. Let us call this method ORF′. The transformation matrix thus has the following form:

$$\mathbf{W}_{\mathrm{ORF'}} = \frac{\sqrt{d}}{\sigma}\mathbf{Q}. \tag{4}$$

**Theorem 2.** *(Appendix B) Let $K_{ORF'}(\mathbf{x}, \mathbf{y})$ be the approximate kernel computed with linear transformation matrix (4). Let $D \leq d$ and $z = ||\mathbf{x} - \mathbf{y}||/\sigma$. There exists a function $f$ such that the bias of $K_{ORF'}(\mathbf{x}, \mathbf{y})$ satisfies*

$$\left|\mathbb{E}(K_{ORF'}(\mathbf{x}, \mathbf{y})) - e^{-z^2/2}\right| \leq e^{-z^2/2}\frac{z^4}{4d} + \frac{f(z)}{d^2},$$

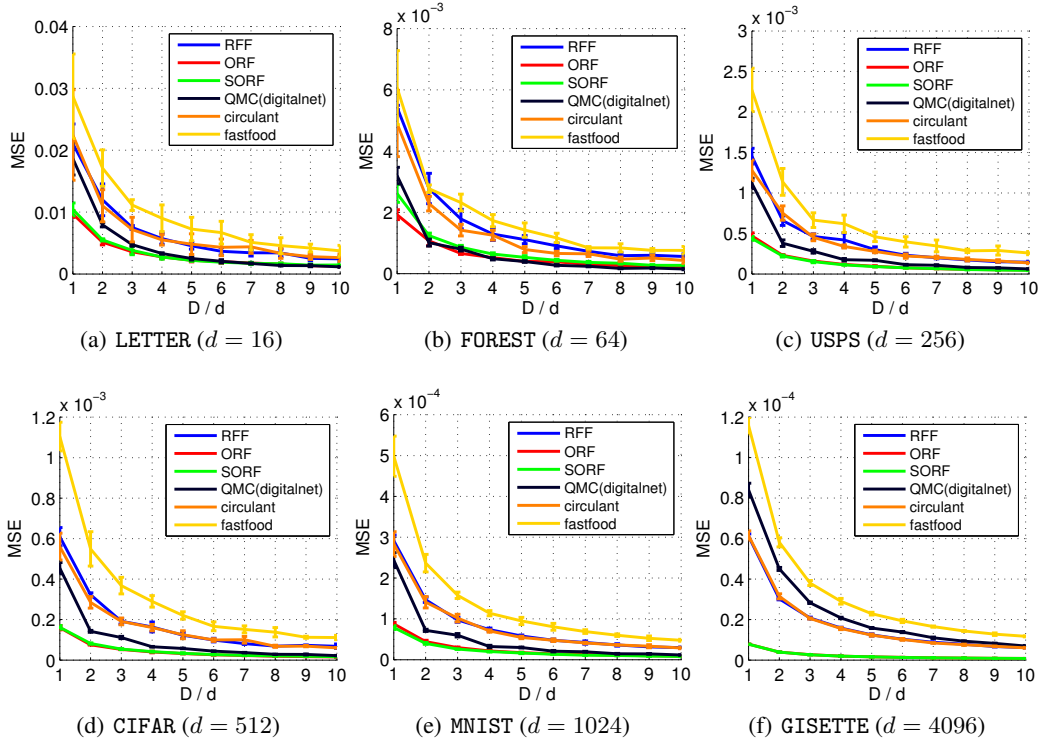

Figure 4: Kernel approximation mean squared error (MSE) for the Gaussian kernel $K(\mathbf{x}, \mathbf{y}) = e^{-||\mathbf{x}-\mathbf{y}||^2/2\sigma^2}$. $D$: number of transformations. $d$: input feature dimension. For each dataset, $\sigma$ is chosen to be the mean distance of the 50th $\ell_2$ nearest neighbor for 1,000 sampled datapoints. Empirically, this yields good classification results. The curves for SORF and ORF overlap.

*and the variance satisfies*

$$Var\left(K_{ORF'}(\mathbf{x}, \mathbf{y})\right) \leq \frac{1}{2D}\left((1 - e^{-z^2})^2 - \frac{D-1}{d}e^{-z^2}z^4\right) + \frac{f(z)}{d^2}.$$

The above implies that when $d$ is large $K_{\text{ORF}'}(\mathbf{x}, \mathbf{y})$ is a good estimation of the kernel with low variance. Figure 3(a) shows that even for relatively small $d$, the estimation is almost unbiased. Figure 3(c) shows that when $d \geq 32$, the variance ratio is very close to that of $d = \infty$. We find empirically that ORF' also provides very similar MSE in comparison with ORF in real-world datasets.

We now introduce Structured Orthogonal Random Features (SORF). It replaces the random orthogonal matrix $\mathbf{Q}$ of ORF' in (4) by a special type of structured matrix $\mathbf{HD}_1\mathbf{HD}_2\mathbf{HD}_3$:

$$\mathbf{W}_{\text{SORF}} = \frac{\sqrt{d}}{\sigma}\mathbf{HD}_1\mathbf{HD}_2\mathbf{HD}_3, \tag{5}$$

where $\mathbf{D}_i \in \mathbb{R}^{d \times d}, i = 1, 2, 3$ are diagonal "sign-flipping" matrices, with each diagonal entry sampled from the Rademacher distribution. $\mathbf{H}$ is the normalized Walsh-Hadamard matrix.

Computing $\mathbf{W}_{\text{SORF}}\mathbf{x}$ has the time cost $\mathcal{O}(d \log d)$, since multiplication with $\mathbf{D}$ takes $\mathcal{O}(d)$ time and multiplication with $\mathbf{H}$ takes $\mathcal{O}(d \log d)$ time using fast Hadamard transformation. The computation of SORF can also be carried out with almost no extra memory due to the fact that both sign flipping and the Walsh-Hadamard transformation can be efficiently implemented as in-place operations [9].

Figures 3(b)(d) show the bias and variance of SORF. Note that although the curves for small $d$ are different from those of ORF, when $d$ is large ($d > 32$ in practice), the kernel estimation is almost unbiased, and the variance ratio converges to that of ORF. In other words, it is clear that SORF can provide almost identical kernel approximation quality as that of ORF. This is also confirmed by the experiments in Section 5. In Section 6, we provide theoretical discussions to show that the structure of (5) can also be generally applied to many scenarios where random Gaussian matrices are used.

| Dataset | Method | $D = 2d$ | $D = 4d$ | $D = 6d$ | $D = 8d$ | $D = 10d$ | Exact |
|---|---|---|---|---|---|---|---|
| letter<br>$d = 16$ | RFF | $76.44 \pm 1.04$ | $81.61 \pm 0.46$ | $\mathbf{85.46 \pm 0.56}$ | $86.58 \pm 0.99$ | $\mathbf{87.84 \pm 0.59}$ | |
| | ORF | $\mathbf{77.49 \pm 0.95}$ | $\mathbf{82.49 \pm 1.16}$ | $85.41 \pm 0.60$ | $\mathbf{87.17 \pm 0.40}$ | $87.73 \pm 0.63$ | $90.10$ |
| | SORF | $76.18 \pm 1.20$ | $81.63 \pm 0.77$ | $84.43 \pm 0.92$ | $85.71 \pm 0.52$ | $86.78 \pm 0.53$ | |
| forest<br>$d = 64$ | RFF | $77.61 \pm 0.23$ | $\mathbf{78.92 \pm 0.30}$ | $79.29 \pm 0.24$ | $79.57 \pm 0.21$ | $\mathbf{79.85 \pm 0.10}$ | |
| | ORF | $\mathbf{77.88 \pm 0.24}$ | $78.71 \pm 0.19$ | $\mathbf{79.38 \pm 0.19}$ | $\mathbf{79.63 \pm 0.21}$ | $79.54 \pm 0.15$ | $80.43$ |
| | SORF | $77.64 \pm 0.20$ | $78.88 \pm 0.14$ | $79.31 \pm 0.12$ | $79.50 \pm 0.14$ | $79.56 \pm 0.09$ | |
| usps<br>$d = 256$ | RFF | $94.27 \pm 0.38$ | $94.98 \pm 0.16$ | $95.43 \pm 0.22$ | $\mathbf{95.66 \pm 0.25}$ | $95.71 \pm 0.18$ | |
| | ORF | $94.21 \pm 0.51$ | $\mathbf{95.26 \pm 0.25}$ | $\mathbf{96.46 \pm 0.18}$ | $95.52 \pm 0.20$ | $\mathbf{95.76 \pm 0.17}$ | $95.57$ |
| | SORF | $\mathbf{94.45 \pm 0.39}$ | $95.20 \pm 0.43$ | $95.51 \pm 0.34$ | $95.46 \pm 0.34$ | $95.67 \pm 0.15$ | |
| cifar<br>$d = 512$ | RFF | $73.19 \pm 0.23$ | $75.06 \pm 0.33$ | $75.85 \pm 0.30$ | $76.28 \pm 0.30$ | $76.54 \pm 0.31$ | |
| | ORF | $\mathbf{73.59 \pm 0.44}$ | $75.06 \pm 0.28$ | $\mathbf{76.00 \pm 0.26}$ | $76.29 \pm 0.26$ | $\mathbf{76.69 \pm 0.09}$ | $78.71$ |
| | SORF | $73.54 \pm 0.26$ | $\mathbf{75.11 \pm 0.21}$ | $75.76 \pm 0.21$ | $\mathbf{76.48 \pm 0.24}$ | $76.47 \pm 0.28$ | |
| mnist<br>$d = 1024$ | RFF | $94.83 \pm 0.13$ | $95.48 \pm 0.10$ | $95.85 \pm 0.07$ | $\mathbf{96.02 \pm 0.06}$ | $95.98 \pm 0.05$ | |
| | ORF | $94.95 \pm 0.25$ | $\mathbf{95.64 \pm 0.06}$ | $\mathbf{95.85 \pm 0.09}$ | $95.95 \pm 0.08$ | $\mathbf{96.06 \pm 0.07}$ | $97.14$ |
| | SORF | $\mathbf{94.98 \pm 0.18}$ | $95.48 \pm 0.08$ | $95.77 \pm 0.09$ | $95.98 \pm 0.05$ | $96.02 \pm 0.07$ | |
| gisette<br>$d = 4096$ | RFF | $\mathbf{97.68 \pm 0.28}$ | $\mathbf{97.74 \pm 0.11}$ | $97.66 \pm 0.25$ | $\mathbf{97.70 \pm 0.16}$ | $\mathbf{97.74 \pm 0.05}$ | |
| | ORF | $97.56 \pm 0.17$ | $97.72 \pm 0.15$ | $\mathbf{97.80 \pm 0.07}$ | $97.64 \pm 0.09$ | $97.68 \pm 0.04$ | $97.60$ |
| | SORF | $97.64 \pm 0.17$ | $97.62 \pm 0.04$ | $97.64 \pm 0.11$ | $97.68 \pm 0.08$ | $97.70 \pm 0.14$ | |

Table 2: Classification Accuracy based on SVM. ORF and SORF provide competitive classification accuracy for a given $D$. Exact is based on kernel-SVM trained on the Gaussian kernel. Note that in all the settings SORF is faster than RFF and ORF by a factor of $\mathcal{O}(d/\log d)$. For example, on gisette with $D = 2d$, SORF provides 10 times speedup in comparison with RFF and ORF.

## 5   Experiments

**Kernel Approximation.** We first show kernel approximation performance on six datasets. The input feature dimension $d$ is set to be power of 2 by padding zeros or subsampling. Figure 4 compares the mean squared error (MSE) of all methods. For fixed $D$, the kernel approximation MSE exhibits the following ordering:

$$\text{SORF} \simeq \text{ORF} < \text{QMC [25]} < \text{RFF [19]} < \text{Other fast kernel approximations [13, 28].}$$

By imposing orthogonality on the linear transformation matrix, Orthogonal Random Features (ORF) achieves significantly lower approximation error than Random Fourier Features (RFF). The Structured Orthogonal Random Features (SORF) have almost identical MSE to that of ORF. All other fast kernel approximation methods, such as circulant [28] and FastFood [13] have higher MSE. We also include DigitalNet, the best performing method among Quasi-Monte Carlo techniques [25]. Its MSE is lower than that of RFF, but still higher than that of ORF and SORF. The order of time cost for a fixed $D$ is

$$\text{SORF} \simeq \text{Other fast kernel approximations [13, 28]} \ll \text{ORF} = \text{QMC [25]} = \text{RFF [19].}$$

Remarkably, SORF has both better computational efficiency and higher kernel approximation quality compared to other methods.

We also apply ORF and SORF on classification tasks. Table 2 shows classification accuracy for different kernel approximation techniques with a (linear) SVM classifier. SORF is competitive with or better than RFF, and has greatly reduced time and space costs.

**The Role of $\sigma$.** Note that a very small $\sigma$ will lead to overfitting, and a very large $\sigma$ provides no discriminative power for classification. Throughout the experiments, $\sigma$ for each dataset is chosen to be the mean distance of the 50th $\ell_2$ nearest neighbor, which empirically yields good classification results [28]. As shown in Section 3, the relative improvement over RFF is positively correlated with $\sigma$. Figure 5(a)(b) verify this on the mnist dataset. Notice that the proposed methods (ORF and SORF) consistently improve over RFF.

**Simplifying SORF.** The SORF transformation consists of three Hadamard-Diagonal blocks. A natural question is whether using fewer computations and randomness can achieve similar empirical performance. Figure 5(c) shows that reducing the number of blocks to two (HDHD) provides similar performance, while reducing to one block (HD) leads to large error.

## 6   Analysis and General Applicability of the Hadamard-Diagonal Structure

We provide theoretical discussions of SORF in this section. We first show that for large $d$, SORF is an unbiased estimator of the Gaussian kernel.

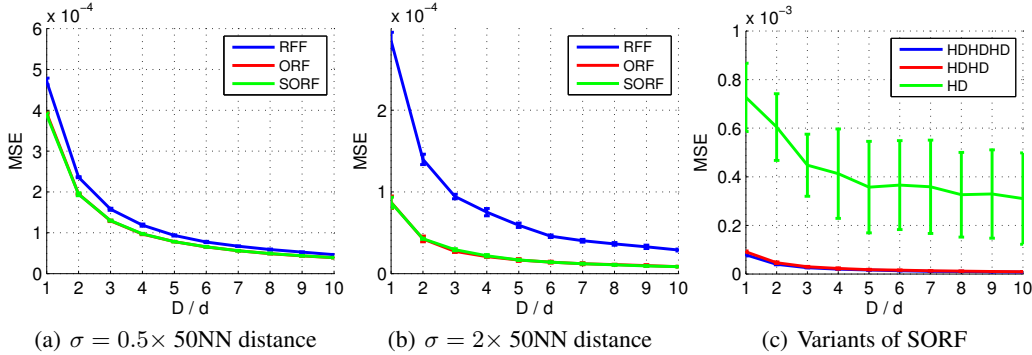

| (a) $\sigma = 0.5\times$ 50NN distance | (b) $\sigma = 2\times$ 50NN distance | (c) Variants of SORF |

Figure 5: (a) (b) MSE on `mnist` with different $\sigma$. (c) Effect of using less randomness on `mnist`. HDHDHD is the the proposed SORF method. HDHD reduces the number of Hadamard-Diagonal blocks to two, and HD uses only one such block.

**Theorem 3.** *(Appendix C) Let $K_{SORF}(\mathbf{x}, \mathbf{y})$ be the approximate kernel computed with linear transformation matrix $\sqrt{d}\mathbf{HD}_1\mathbf{HD}_2\mathbf{HD}_3$. Let $z = ||\mathbf{x} - \mathbf{y}||/\sigma$. Then*

$$\left| \mathbb{E}(K_{SORF}(\mathbf{x}, \mathbf{y})) - e^{-z^2/2} \right| \leq \frac{6z}{\sqrt{d}}.$$

Even though SORF is nearly-unbiased, proving tight variance and concentration guarantees similar to ORF remains an open question. The following discussion provides a sketch in that direction. We first show a lemma of RFF.

**Lemma 3.** *Let $\mathbf{W}$ be a random Gaussian matrix as in RFF, for a given $\mathbf{z}$, the distribution of $\mathbf{Wz}$ is $N(0, ||z||_2\mathbf{I}_d)$.*

Note that $\mathbf{Wz}$ in RFF can be written as $\mathbf{Rg}$, where $\mathbf{R}$ is a scaled orthogonal matrix such that each row has norm $||z||_2$ and $\mathbf{g}$ is distributed according to $N(0, \mathbf{I}_d)$. Hence the distribution of $\mathbf{Rg}$ is $N(0, ||z||_2\mathbf{I}_d)$, identical to $\mathbf{Wz}$. The concentration results of RFF use the fact that the projections of a Gaussian vector $\mathbf{g}$ onto orthogonal directions $\mathbf{R}$ are independent.

We show that $\sqrt{d}\mathbf{HD}_1\mathbf{HD}_2\mathbf{HD}_3\mathbf{z}$ has similar properties. In particular, we show that it can be written as $\tilde{\mathbf{R}}\tilde{\mathbf{g}}$, where rows of $\tilde{\mathbf{R}}$ are "near-orthogonal" (with high probability) and have norm $||\mathbf{z}||_2$, and the vector $\tilde{\mathbf{g}}$ is close to Gaussian ($\tilde{\mathbf{g}}$ has independent sub-Gaussian elements), and hence the projections behave "near-independently". Specifically, $\tilde{\mathbf{g}} = vec(\mathbf{D}_1)$ (vector of diagonal entries of $\mathbf{D}_1$), and $\tilde{\mathbf{R}}$ is a function of $\mathbf{D}_2$, $\mathbf{D}_3$ and $\mathbf{z}$.

**Theorem 4.** *(Appendix D) For a given $\mathbf{z}$, there exists a $\tilde{\mathbf{R}}$ (function of $\mathbf{D}_2, \mathbf{D}_3, \mathbf{z}$), such that $\sqrt{d}\mathbf{HD}_1\mathbf{HD}_2\mathbf{HD}_3\mathbf{z} = \tilde{\mathbf{R}}vec(\mathbf{D}_1)$. Each row of $\tilde{\mathbf{R}}$ has norm $||\mathbf{z}||_2$ and for any $t \geq 1/d$, with probability $1 - de^{-c \cdot t^{2/3}d^{1/3}}$, the inner product between any two rows of $\tilde{\mathbf{R}}$ is at most $t||z||_2$, where $c$ is a constant.*

The above result can also be applied to settings not limited to kernel approximation. In the appendix, we show empirically that the same scheme can be successfully applied to angle estimation where the nonlinear map $f$ is a non-smooth $\text{sign}(\cdot)$ function [4]. We note that the $\mathbf{HD}_1\mathbf{HD}_2\mathbf{HD}_3$ structure has also been recently used in fast cross-polytope LSH [2, 12, 6].

## 7  Conclusions

We have demonstrated that imposing orthogonality on the transformation matrix can greatly reduce the kernel approximation MSE of Random Fourier Features when approximating Gaussian kernels. We further proposed a type of structured orthogonal matrices with substantially lower computation and memory cost. We provided theoretical insights indicating that the Hadamard-Diagonal block structure can be generally used to replace random Gaussian matrices in a broader range of applications. Our method can also be generalized to other types of kernels such as general shift-invariant kernels and polynomial kernels based on Schoenberg's characterization as in [18].

## Footnotes

[1] We first generate the random Gaussian matrix $\mathbf{G}$ in (1). $\mathbf{Q}$ is the orthogonal matrix obtained from the QR decomposition of $\mathbf{G}$. $\mathbf{Q}$ is distributed uniformly on the Stiefel manifold (the space of all orthogonal matrices) based on the Bartlett decomposition theorem [16].

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
