[Supplementary Material · supp.pdf]

# Supplementary material of Orthogonal Random Features

## Appendix A    Variance Reduction via Orthogonal Random Features

### A.1    Notation

Let $\mathbf{z} = \frac{\mathbf{x}-\mathbf{y}}{\sigma}$, and $z = ||\mathbf{z}||$. For a vector $\mathbf{y}$, let $y(i)$ denote its $i^{\text{th}}$ coordinate. Let $n!!$ be the double factorial of $n$, i.e., the product of every number from n to 1 that has the same parity as n.

### A.2    Proof of Lemma 1

Let $\mathbf{z} = (\mathbf{x} - \mathbf{y})/\sigma$. Recall that in RFF, we compute the Kernel approximation as

$$\sum_{i=1}^{D} \frac{1}{D} \cos(\mathbf{w}_i^T \mathbf{z}),$$

where each $\mathbf{w}_i$ is a $d$ dimensional vector distributed $N(0, I_d)$. Let $\mathbf{w}$ be a $d$ dimensional vector distributed $N(0, I_d)$. By Bochner's theorem,

$$\mathbb{E}[\cos(\mathbf{w}^T \mathbf{z})] = e^{-z^2/2},$$

and hence RFF yields an unbiased estimate.

We now compute the variance of RFF approximation. Observe that

$$\cos^2(\mathbf{w}^T \mathbf{z}) = \frac{1 + \cos(2\mathbf{w}^T \mathbf{z})}{2} = \frac{1 + \cos(\mathbf{w}^T(2\mathbf{z}))}{2}.$$

Hence by Bochner's theorem

$$\mathbb{E}[\cos^2(\mathbf{w}^T \mathbf{z})] = \frac{1 + e^{-2z^2}}{2}.$$

Therefore,

$$\mathrm{Var}(\cos(\mathbf{w}^T \mathbf{z})) = \mathbb{E}[\cos^2(\mathbf{w}^T \mathbf{z})] - (\mathbb{E}[\cos(\mathbf{w}^T \mathbf{z})])^2$$

$$= \frac{1 + e^{-2z^2}}{2} - e^{-z^2} = \frac{(1 - e^{-z^2})^2}{2}.$$

If we take $D$ such independent random variables $\mathbf{w}_1, \mathbf{w}_2, \ldots \mathbf{w}_D$, since variance of the sum is sum of variances,

$$\mathrm{Var}\left(\frac{1}{D}\sum_{i=1}^{D}\cos(\mathbf{w}_i^T\mathbf{z})\right) = \frac{(1 - e^{-z^2})^2}{2D}.$$

### A.3    Proof of Lemma 2

The proof uses the following lemma.

**Lemma 4.** *For a set of non-negative values $\alpha_1, \alpha_2, \ldots \alpha_k$ and $\beta_1, \beta_2, \ldots \beta_k$ such that for all $i$, $\beta_i \leq \alpha_i$,*

$$\left| \frac{1}{\prod_i^k (1+\alpha_i)} - \left(1 - \sum_{i=1}^{k}\alpha_i\right)\right| \leq \left(\sum_{i=1}^{k}\alpha_i\right)^2,$$

*and*

$$\left| \prod_i^k \frac{1+\beta_i}{1+\alpha_i} - \left(1 + \sum_{i=1}^{k}\beta_i - \sum_{i=1}^{k}\alpha_i\right)\right| \leq \left(\sum_{i=1}^{k}(\alpha_i - \beta_i)\right)^2 + \sum_{i=1}^{k}(\alpha_i - \beta_i)\beta_i.$$

*Proof.* Since $\alpha_i$s are non-negative,

$$\frac{1}{\prod_i^k(1+\alpha_i)} - \left(1 - \sum_{i=1}^k \alpha_i\right) \leq \frac{1}{1+\sum_i^k \alpha_i} - \left(1 - \sum_{i=1}^k \alpha_i\right)$$

$$= \frac{1 - (1+\sum_{i=1}^k \alpha_i)(1 - \sum_{i=1}^k \alpha_i)}{1+\sum_i^k \alpha_i}$$

$$= \frac{(\sum_{i=1}^k \alpha_i)^2}{1+\sum_i^k \alpha_i} \leq \left(\sum_{i=1}^k \alpha_i\right)^2.$$

Furthermore, by convexity

$$\frac{1}{\prod_i^k(1+\alpha_i)} \geq \frac{1}{(1+\sum_{i=1}^k \alpha_i/k)^k} \geq e^{-\sum_{i=1}^k \alpha_i} \geq 1 - \sum_{i=1}^k \alpha_i.$$

Combining the above two equations results in the first part of the lemma. For the second part observe that

$$\prod_i^k \frac{1+\beta_i}{1+\alpha_i} = \frac{1}{\prod_i^k \left(1 + \frac{\alpha_i-\beta_i}{1+\beta_i}\right)}.$$

Hence, by the first part

$$\left| \prod_i^k \frac{1+\beta_i}{1+\alpha_i} - \left(1 - \sum_{i=1}^k \frac{\alpha_i - \beta_i}{1+\beta_i}\right) \right| \leq \left(\sum_{i=1}^k \frac{\alpha_i - \beta_i}{1+\beta_i}\right)^2 \leq \left(\sum_{i=1}^k (\alpha_i - \beta_i)\right)^2.$$

Furthermore, for every $i$

$$\left| \frac{1}{1+\beta_i} - 1 \right| \leq \beta_i.$$

Combining the above two equations yields the second part of the lemma. $\qquad\square$

*Proof of Lemma 2.* Observe that

$$\cos(\mathbf{w}_1^T \mathbf{z}) \cos(\mathbf{w}_2^T \mathbf{z}) = \frac{cos(\mathbf{w}_1^T \mathbf{z} + \mathbf{w}_2^T \mathbf{z}) + \cos(\mathbf{w}_1^T \mathbf{z} - \mathbf{w}_2^T \mathbf{z})}{2}.$$

Since the problem is rotation invariant, instead of projecting a vector $\mathbf{z}$ onto a randomly chosen two orthogonal vectors $\mathbf{u}_1$ and $\mathbf{u}_2$, we can choose a vector $\mathbf{y}$ that is uniformly distributed on a sphere of radius $z$ and project it on to the first two dimensions. Thus,

$$\mathbb{E}[\cos(\mathbf{w}_1^T \mathbf{z} + \mathbf{w}_2^T \mathbf{z})] = \mathbb{E}[\cos((s_1 y(1) + s_2 y(2))z)].$$

Similarly,

$$\mathbb{E}[\cos(\mathbf{w}_1^T \mathbf{z} - \mathbf{w}_2^T \mathbf{z})] = \mathbb{E}[\cos((s_1 y(1) - s_2 y(2))z)].$$

The $k^{\text{th}}$ term in the Taylor's series expansion of sum of above two terms is

$$\frac{(-1)^k}{(2k)!} ((s_1 y(1) + s_2 y(2))z)^{2k} + \frac{(-1)^k}{(2k)!} ((s_1 y(1) - s_2 y(2))z)^{2k}$$

$$= \frac{(-z^2)^k}{(2k)!} \sum_{i=0}^k \binom{2k}{2i} s_1^{2i} y^{2i}(1) s_2^{2k-2i} y^{2k-2i}(2).$$

A way to compute a uniformly distributed random variable on a sphere with radius $z$ is to generate $d$ independent random variables $\mathbf{x} = (x(1), x(2), \ldots, x(d))$ each distributed $N(0,1)$ and setting

$y(i) = zx(i)/||\mathbf{x}||$. Hence,

$$\mathbb{E}\left[\sum_{i=0}^{k}\binom{2k}{2i}s_1^{2i}y^{2i}(1)s_2^{2k-2i}y^{2k-2i}(2)\right]$$

$$\stackrel{(a)}{=}\mathbb{E}\left[\sum_{i=0}^{2k}\binom{2k}{2i}\frac{s_1^{2i}x^{2i}(1)s_2^{2k-2i}x^{2k-2i}(2)}{||\mathbf{x}||^{2k}}\right]$$

$$\stackrel{(b)}{=}\sum_{i=0}^{k}\binom{2k}{2i}\mathbb{E}[s_1^{2i}]\mathbb{E}[s_2^{2k-2i}]\mathbb{E}\left[\frac{x^{2i}(1)x^{2k-2i}(2)}{||\mathbf{x}||^{2k}}\right]$$

$$\stackrel{(c)}{=}\sum_{i=0}^{k}\binom{2k}{2i}\mathbb{E}[s_1^{2i}]\mathbb{E}[s_2^{2k-2i}]\frac{\mathbb{E}[x^{2i}(1)]\mathbb{E}[x^{2k-2i}(2)]}{\mathbb{E}[||\mathbf{x}||^{2k}]}$$

$$\stackrel{(d)}{=}\sum_{i=0}^{k}\binom{2k}{2i}\frac{(d+2i-2)!!(d+2k-2i-2)!!\cdot(2i-1)!!(2k-2i-1)!!}{(d+2k-2)!!(d-2)!!}$$

$$\stackrel{(e)}{=}\frac{(2k)!}{2^k k!}\sum_{i=0}^{k}\binom{k}{i}\frac{(d+2i-2)!!(d+2k-2i-2)!!}{(d+2k-2)!!(d-2)!!}.$$

$(a)$ follows from linearity of expectation and the observation above. $(b)$ follows from the independence of $s_1$, $s_2$, and $\mathbf{x}$. $(d)$ follows from substituting the moments of chi and Gaussian distributions. $(e)$ follows from numerical simplification. We now describe the reasoning behind $(c)$. Let $\mathbf{z} = \frac{\mathbf{x}||\mathbf{y}||}{||\mathbf{x}||}$, where $\mathbf{y}$ and $\mathbf{x}$ are independent $N(0, I_d)$ random variables. By the properties of the Gaussian random variables $\mathbf{z}$ is also a $N(0, I_d)$ random variable. Thus,

$$\mathbb{E}[z^{2i}(1)]\mathbb{E}[z^{2k-2i}(2)] = \mathbb{E}\left[\frac{x^{2i}(1)x^{2k-2i}(2)}{||\mathbf{x}||^{2k}}\right]\mathbb{E}[||\mathbf{y}||^{2k}].$$

Rearranging terms, we get

$$\mathbb{E}\left[\frac{x^{2i}(1)x^{2k-2i}(2)}{||\mathbf{x}||^{2k}}\right] = \frac{\mathbb{E}[z^{2i}(1)]\mathbb{E}[z^{2k-2i}(2)]}{\mathbb{E}[||\mathbf{y}||^{2k}]} = \frac{\mathbb{E}[x^{2i}(1)]\mathbb{E}[x^{2k-2i}(2)]}{\mathbb{E}[||\mathbf{x}||^{2k}]},$$

and hence $(c)$. Substituting the above equation in the cosine expansion, we get that the expectation is

$$\mathbb{E}[\cos(s_1 y(1) + s_2 y(2))] = \sum_{k=0}^{\infty}\frac{(-z^2)^k}{k!}\sum_{i=0}^{k}\binom{k}{i}\frac{1}{2^k}\frac{(d+2i-2)!!(d+2k-2i-2)!!}{(d+2k-2)!!(d-2)!!}.$$

Observe that

$$\frac{(d+2i-2)!!(d+2k-2i-2)!!}{(d+2k-2)!!(d-2)!!} = \frac{\prod_{j=0}^{k-i-1}(1+2j/d)}{\prod_{j=0}^{k-i-1}(1+2(j+i)/d)},$$

Hence by Lemma 4,

$$\left|\frac{\prod_{j=0}^{k-i-1}(1+2j/d)}{\prod_{j=0}^{k-i-1}(1+2(j+i)/d)} - \left(1 + \sum_{j=0}^{k-i-1}\frac{2j}{d} - \sum_{j=0}^{k-i-1}\frac{2(j+i)}{d}\right)\right|$$

$$\leq \left(\sum_{j=0}^{k-i-1}\frac{2i}{d}\right)^2 + \sum_{j=0}^{k-i-1}\frac{2i}{d}\left(\frac{2j}{d}\right).$$

Simplifying we get,

$$\left|\frac{\prod_{j=0}^{k-i-1}(1+2j/d)}{\prod_{j=0}^{k-i-1}(1+2(j+i)/d)} - \left(1 + \frac{2i^2 - 2ik}{d}\right)\right| \leq \frac{4i^2(k-i)^2}{d^2} + \frac{2i(k-i)(k-i-1)}{d^3}.$$

Hence summing over $i$,

$$\left| \sum_{i=0}^{k} \binom{k}{i} \frac{1}{2^k} \frac{\prod_{j=0}^{k-i-1}(1+2j/d)}{\prod_{j=0}^{k-i-1}(1+2(j+i)/d)} - \left(1 + \frac{k-k^2}{2d}\right) \right| \le \frac{k^4}{4d^2} + \frac{k^2(k-1)}{2d^3}.$$

Substituting,

$$\frac{\mathbb{E}[\cos((s_1y(1)+s_2y(2))z]] + \mathbb{E}[\cos((s_1y(1)-s_2y(2))z]]}{2} = \sum_{k=0}^{\infty} \frac{(-z^2)^k}{k!} \left(1 + \frac{k-k^2}{2d} + c_{k,d}\right),$$

where $|c_{k,d}| \le \frac{k^4}{4d^2} + \frac{k^2(k-1)}{2d^3}$. Thus,

$$\frac{\mathbb{E}[\cos((s_1y(1)+s_2y(2))z]] + \mathbb{E}[\cos((s_1y(1)-s_2y(2))z]]}{2}$$

$$= \sum_{k=0}^{\infty} \frac{(-z^2)^k}{k!} \left(1 + \frac{k-k^2}{2d} + c_{k,d}\right)$$

$$\le \sum_{k=0}^{\infty} \frac{(-z^2)^k}{k!} \left(1 + \frac{k-k^2}{2d}\right) + \sum_{k=0}^{\infty} \frac{(z^2)^k}{k!} \left(\frac{k^4}{4d^2} + \frac{k^2(k-1)}{2d^3}\right)$$

$$\le e^{-z^2} - e^{-z^2} \frac{z^4}{2d} + \frac{e^{z^2}(z^8 + 6z^6 + 7z^4 + z^2)}{4d^2} + \frac{e^{z^2} z^4(z^6 + 2z^4)}{2d^3}.$$

<div style="text-align:right">□</div>

## Appendix B    Proof of Theorem 2

The proof of the theorem is similar to that of Lemma 2 and we outline some key steps. We first bound the bias in Lemma 5 and then the variance in Lemma 6.

**Lemma 5.** *If* $\mathbf{w} = \sqrt{d}\mathbf{y}$, *where* $y$ *is distributed uniformly on a unit sphere, then*

$$\left| \mathbb{E}[\cos \mathbf{w}^T \mathbf{z}] - \left(e^{-z^2/2} - e^{-z^2/2} \frac{z^4}{4d}\right) \right| \le \frac{e^{z^2/2} z^4(z^4 + 8z^2 + 8)}{16d^2}.$$

*Proof.* Without loss of generality, we can assume $\mathbf{z}$ is along the first coordinate and hence $\mathbf{w}^T\mathbf{z} = \sqrt{d}zy(1)$. A way to compute a uniformly distributed random variable on a sphere with radius $z$ is to generate $d$ independent random variables $\mathbf{x} = (x(1), x(2), \ldots, x(d))$ each distributed $N(0,1)$ and setting $y(i) = zx(i)/||\mathbf{x}||$. hence,

$$\mathbb{E}[\cos \mathbf{w}^T \mathbf{z}] = \mathbb{E}\left[\cos\left(\frac{z\sqrt{d}x(1)}{||\mathbf{x}||}\right)\right].$$

The $k^{\text{th}}$ term in the Taylor's series expansion of cosine in the above equation is

$$\frac{(-1)^k}{(2k)!} \left(\frac{\sqrt{d}x(1)z}{||\mathbf{x}||}\right)^{2k}$$

Similar to the proof of Lemma 2, it can be shown that the expectation of this term is

$$\mathbb{E}\left[\frac{(-1)^k}{(2k)!}\left(\frac{z\sqrt{d}x(1)}{||\mathbf{x}||}\right)^{2k}\right] = \frac{(-z^2)^k}{2^k k!} \frac{d^k}{(d, 2k-2)!!}.$$

Applying Lemma 4 and simplifying,

$$\mathbb{E}[\cos(dy(1))] = \sum_{k=0}^{\infty} \frac{(-z^2)^k}{2^k k!} \left(1 - \frac{k(k-1)}{d} + c'_{k,d}\right),$$

where $|c'_{k,d}| \leq \left(\frac{k(k-1)}{d}\right)^2$. Hence,

$$\left| \mathbb{E}[\cos(dy(1))] - \sum_{k=0}^{\infty} \frac{(-z^2)^k}{2^k k!} \left(1 - \frac{k(k-1)}{d}\right) \right| \leq \sum_{k=0}^{\infty} \frac{(z^2)^k}{2^k k!} \left(\frac{k(k-1)}{d}\right)^2,$$

and thus

$$\left| \mathbb{E}[\cos(dy(1))] - e^{-z^2/2} + e^{-z^2/2} \frac{z^4}{4d} \right| \leq \frac{e^{z^2/2} z^4 (z^4 + 8z^2 + 8)}{16d^2}.$$

$\square$

**Lemma 6.** *Let $D \leq d$. If $\mathbf{W} = \sqrt{d}\mathbf{Q}$, where $\mathbf{Q}$ is a uniformly chosen random rotation, then*

$$Var\left(\frac{1}{D} \sum_{i=1}^{D} \cos(\mathbf{w}_i^T \mathbf{z})\right) \leq \frac{1}{2D}\left((1 - e^{-z^2})^2 - \frac{D-1}{d} e^{-z^2} z^4\right) + \frac{\mathcal{O}(e^{3z^2})}{d^2}.$$

*Proof.* Let $a_i = \cos(\mathbf{w}_i^T \mathbf{z})$. Expanding the variance we have,

$$\mathrm{Var}\left(\frac{1}{D}\sum_{i=1}^{D} a_i\right) = \frac{1}{D^2}\sum_i \left(\mathbb{E}[a_i^2] - (\mathbb{E}[a_i])^2\right) + \frac{1}{D^2}\sum_i \sum_{j \neq i} \left(\mathbb{E}[a_i a_j] - \mathbb{E}[a_i]\mathbb{E}[a_j]\right)$$

$$= \frac{1}{D}\left(\mathbb{E}[a_1^2] - (\mathbb{E}[a_1]^2)\right) + \frac{D-1}{D}\left(\mathbb{E}[a_1 a_2] - \mathbb{E}[a_1]\mathbb{E}[a_2]\right).$$

For the first term, rewriting $\cos^2(\mathbf{w}^T \mathbf{z}) = \frac{1+\cos(2\mathbf{w}^T \mathbf{z})}{2}$, similar to the proof of Lemma 5 it can be shown that

$$(\mathbb{E}[a_1^2] - (\mathbb{E}[a_1]^2) \leq \frac{(1 - e^{-z^2})^2}{2} + \frac{\mathcal{O}(e^{3z^2})}{d}.$$

Second term can be bounded similar to Lemma 2 and here we just sketch an outline. Similar to the proof of Lemma 2, the variance boils down to computing the expectation of $\cos(\mathbf{w}_1^T \mathbf{z} + \mathbf{w}_2^T \mathbf{z})$. Using Lemma 4 and summing Taylor's series we get

$$\left| \mathbb{E}[\cos(\mathbf{w}_1^T \mathbf{z} + \mathbf{w}_2^T \mathbf{z})] - e^{-z^2} + e^{-z^2} \frac{z^4}{d} \right| \leq \frac{e^{z^2} z^4 (z^4 + 4z^2 + 2)}{d^2}.$$

Substituting the above bound and the expectation from Lemma 5, we get

$$\mathbb{E}[a_1 a_2] - \mathbb{E}[a_1]\mathbb{E}[a_2] \leq -e^{z^2} \frac{z^4}{2d} + \frac{\mathcal{O}(e^{3z^2})}{d^2},$$

and hence the lemma. $\square$

## Appendix C   Proof of Theorem 3

The proof follows from the following two technical lemmas.

**Lemma 7.** *Let $z'$ be distributed according to $N(0, ||\mathbf{x}||_2^2)$ and $y' = \sum_{i=1}^{d} x(i)d_i$, where $d_i$s are independent Rademacher random variables. For any function $g$ such that $|g'| \leq 1$ and $|g| \leq 1$,*

$$|\mathbb{E}[g(z')] - \mathbb{E}[g(y')]| \leq \frac{3}{2} \sum_{i=1}^{d} \frac{x^3(i)}{||\mathbf{x}||_2^2}.$$

*Proof.* Let $z = z'/||\mathbf{x}||_2$, $y = y'/||\mathbf{x}||_2$, and $h(x) = g(||\mathbf{x}||_2 x)$, for all $x$. Hence $h(z) = g(z')$ and $h(y) = g(y')$. By a lemma due to Stein [30],

$$|\mathbb{E}[g(z')] - \mathbb{E}[g(y')]| = |\mathbb{E}[h(z)] - \mathbb{E}[h(y)]|$$

$$\leq \sup_f \{|\mathbb{E}[f'(y) - yf(y)]| : |f|_\infty \leq ||\mathbf{x}||_2, |f'|_\infty \leq \sqrt{2/\pi}||\mathbf{x}||_2, |f''|_\infty \leq 2||\mathbf{x}||_2\}.$$

We now bound the term on the right hand side by classic Stein-type arguments.

$$\mathbb{E}[yf(y)] = \sum_{i=1}^{d} \frac{x(i)d_i}{||\mathbf{x}||_2} \mathbb{E}[f(y)].$$

Let $y_i = y - \frac{x(i)d_i}{||\mathbf{x}||_2}$. Observe that

$$\mathbb{E}[d_i f(y)] = \mathbb{E}[d_i(f(y) - f(y_i))]$$
$$= \mathbb{E}[d_i(f(y) - f(y_i)) - d_i(y - y_i)f'(y_i)] + \mathbb{E}[d_i(y - y_i)f'(y_i)],$$

where the first equality follows from the fact that $y_i$ and $d_i$ are independent and $d_i$ has zero mean. By Taylor series approximation, the first term is bounded by

$$|\mathbb{E}[d_i f(y) - f(y_i) - d_i(y - y_i)f'(y_i)]| \leq \frac{1}{2}(y - y_i)^2 |f''|_\infty = \frac{1}{2}\frac{x^2(i)}{||\mathbf{x}||_2^2}|f''|_\infty.$$

Similarly,

$$\mathbb{E}[d_i(y - y_i)f'(y_i)] = \frac{x(i)}{||\mathbf{x}||_2}f'(y_i).$$

Combining the above four equations, we get

$$\left|\mathbb{E}\left[yf(y) - \sum_{i=1}^{d} \frac{x^2(i)}{||\mathbf{x}||_2^2}f'(y_i)\right]\right| \leq \sum_{i=1}^{d} \frac{|x^3(i)|}{||\mathbf{x}||_2^3}|f''|_\infty.$$

Similarly, note that

$$\left|\mathbb{E}\left[f'(y) - \sum_{i=1}^{d} \frac{x^2(i)}{||\mathbf{x}||_2^2}f'(y_i)\right]\right| \leq \sum_{i=1}^{d}|f''|_\infty \frac{x^2(i)}{||\mathbf{x}||_2^2}\mathbb{E}[|y - y_i|] = \sum_{i=1}^{d}|f''|_\infty \frac{|x^3(i)|}{||\mathbf{x}||_2^3}.$$

Combining the above two equations, we get

$$||\mathbb{E}[yf(y) - f'(y))]| \leq \frac{3|f''|_\infty}{2}\sum_{i=1}^{d}\frac{|x^3(i)|}{||\mathbf{x}||_2^3}.$$

Substituting the bound on the second moment of $f$ yields the result. □

Let $\mathbf{G}$ be a random matrix with i.i.d. $N(0, 1)$ entries as before. Using the above lemma we show that $\sqrt{d}\mathbf{H}\mathbf{D}_1\mathbf{H}\mathbf{D}_2$ behaves like $\mathbf{G}$ while computing the bias.

**Lemma 8.** *For a given* $\mathbf{x}$, *let* $\mathbf{z} = \mathbf{G}\mathbf{x}$ *and* $\mathbf{y} = \sqrt{d}\mathbf{H}\mathbf{D}_1\mathbf{H}\mathbf{D}_2\mathbf{x}$. *For any function* $g$ *such that* $|g'| \leq 1$ *and* $|g| \leq 1$,

$$\left|\frac{1}{d}\sum_{i=1}^{d}\mathbb{E}[g(z(i))] - \frac{1}{d}\sum_{i=1}^{d}\mathbb{E}[g(y(i))]\right| \leq 6\frac{||\mathbf{x}||_2}{\sqrt{d}}.$$

*Proof.* By triangle inequality,

$$\left|\frac{1}{d}\sum_{i=1}^{d}\mathbb{E}[g(z(i))] - \frac{1}{d}\sum_{i=1}^{d}\mathbb{E}[g(y(i))]\right| \leq \frac{1}{d}\sum_{i=1}^{d}|\mathbb{E}[g(z(i))] - \mathbb{E}[g(y(i))]|.$$

Let $\mathbf{u} = \mathbf{H}\mathbf{D}_2\mathbf{x}$. Then for every $i$, $y(i) = \sum_j H(i, j)D_2(j)u(j)$. Hence by Lemma 7, we can relate expectation under $y$ to the expectation under Gaussian distribution:

$$|\mathbb{E}[g(z(i))] - \mathbb{E}[g(y(i))]| \overset{(a)}{=} |\mathbb{E}[\mathbb{E}[g(z(i))] - \mathbb{E}[g(y(i))|\mathbf{u}]]|$$

$$\leq \frac{3}{2}\sum_{i=1}^{d}\mathbb{E}\left[\frac{|u^3(i)|}{||\mathbf{u}||_2^2}\right]$$

$$= \frac{3}{2}\sum_{i=1}^{d}\mathbb{E}\left[\frac{|u^3(i)|}{||\mathbf{x}||_2^2}\right],$$

where the last equality follows from the fact that $\mathbf{HD}_2$ does not change rotation and $(a)$ follows from the law of total expectation. By Cauchy-Schwartz inequality, for each $i$

$$\mathbb{E}[|u(i)|^3] \leq \sqrt{\mathbb{E}[u^6(i)]},$$

It can be shown that

$$\mathbb{E}[u^6(i)] \leq \frac{15\|\mathbf{x}\|_2^6}{d^3},$$

Summing over all the indices yields the lemma. □

Theorem 3 follows from the Bochner's theorem and the fact that $\cos(\cdot)$ satisfies requirements for the above lemma. We note that Theorem 3 holds for the matrix $\sqrt{d}\mathbf{HD}_1\mathbf{HD}_2$ itself and the third component $\mathbf{HD}_3$ is not necessary to bound the bias.

## Appendix D   Proof of Theorem 4

To prove Theorem 4, we use the Hanson-Wright Inequality.

**Lemma 9** (Hanson-Wright Inequality). *Let $X = (X_1, ..., X_n) \in \mathbb{R}^n$ be a random vector with independent subgaussian components $X_i$ which satisfy: $\mathbb{E}[X_i] = 0$ and $\|X_i\|_{sg} \leq K$ for some constant $K > 0$. Let $A \in \mathbb{R}^{n \times n}$. Then for any $t > 0$ the following holds:*

$$\mathbb{P}[|X^T A X - \mathbb{E}[X^T A X]| > t] \leq 2e^{-c\min(\frac{t^2}{K^4\|A\|_F^2}, \frac{t}{K^2\|A\|_2})},$$

*for some universal positive constant $c > 0$.*

*Proof of Theorem 4.* For a vector $\mathbf{u}$, let $\text{diag}(\mathbf{u})$ denote the diagonal matrix whose entries correspond to the entries of $\mathbf{u}$. For a diagonal matrix $\mathbf{D}$, let $\text{vec}(\mathbf{D})$ denote the vector corresponding to the diagonal entries of $\mathbf{D}$. Let $\mathbf{v} = \mathbf{HD}_3\mathbf{z}$ and $\mathbf{u} = \mathbf{HD}_2\mathbf{v} = \mathbf{H}\text{diag}(\mathbf{v})\text{vec}(\mathbf{D}_2)$. Observe that

$$\sqrt{d}\mathbf{HD}_1\mathbf{HD}_2\mathbf{HD}_3\mathbf{z} = \sqrt{d}\mathbf{H}\text{diag}(\mathbf{HD}_2\mathbf{HD}_3\mathbf{z})\text{vec}(\mathbf{D}_1).$$

Hence $\tilde{\mathbf{R}} = \sqrt{d}\mathbf{H}\text{diag}(\mathbf{HD}_2\mathbf{HD}_3\mathbf{z})$. Note that all the entries of $\sqrt{d}\mathbf{H}$ have magnitude 1 and $\mathbf{HD}_2\mathbf{HD}_3$ do not change norm of the vector. Hence, each row of $\tilde{\mathbf{R}}$ has norm $\|\mathbf{z}\|_2$. To prove the orthogonality of rows of $\tilde{\mathbf{R}}$, we need to show that for any $i$ and $j \neq i$,

$$\sqrt{d}\sum_{k=1}^d H(i,k)H(j,k)u^2(k)$$

is small. We first show that the expectation of the above quantity is 0 and then use the Hanson-Wright inequality to prove concentration. Let $\mathbf{A}$ be a diagonal matrix with $k^{th}$ entry being $\sqrt{d}H(i,k)H(j,k)$. The above equation can be rewriten as

$$\sum_{k=1}^d H(i,k)H(j,k)u^2(k) = \text{vec}(\mathbf{D}_2)^T\text{diag}(\mathbf{v})\mathbf{H}^T\mathbf{A}\mathbf{H}\text{diag}(\mathbf{v})\text{vec}(\mathbf{D}_2).$$

Observe that the $(l,l)$ entry of the $\mathbf{H}^T\mathbf{A}\mathbf{H}$ is

$$\sum_{k=1}^d H^T(l,k)A(k,k)H(k,l) = \sum_{k=1}^d H(k,l)A(k,k)H(k,l)$$

$$= \frac{1}{d}\sum_{k=1}^d A(k,k)$$

$$= \sum_{k=1}^d H(i,k)H(j,k) = 0,$$

Figure 6: Recall and angular MSE on a 16384-dimensional dataset of natural images [29].

where the last equality follows from observing that the rows of $\mathbf{H}$ are orthogonal to each other. Together with the fact that elements of $\mathbf{D}_2$ are independent of each other, we get

$$\mathbb{E}[\mathbf{u}^T\mathbf{A}\mathbf{u}] = \mathbb{E}[\text{vec}(\mathbf{D}_2)^T\text{diag}(\mathbf{v})\mathbf{H}^T\mathbf{A}\mathbf{H}\text{diag}(\mathbf{v})\text{vec}(\mathbf{D}_2)] = 0,$$

To prove the concentration result, observe that the entries of $\text{vec}(\mathbf{D}_2)$ are independent and sub-Gaussian, and hence we can use the Hanson-Wright inequality. To this end, we bound the Frobenius and the spectral norm of the underlying matrix. For the Frobenius norm, observe that

$$||\text{diag}(\mathbf{v})\mathbf{H}^T\mathbf{A}\mathbf{H}\text{diag}(\mathbf{v})||_F \overset{(a)}{\leq} (||\mathbf{v}||_\infty)^4 ||\mathbf{H}^T\mathbf{A}\mathbf{H}||_F$$
$$\overset{(b)}{=} (||\mathbf{v}||_\infty)^4 ||\mathbf{A}||_F$$
$$\overset{(c)}{=} d(||\mathbf{v}||_\infty)^4,$$

where $(a)$ follows by observing that each $\text{diag}(\mathbf{v})$ changes the Frobenius norm by at most $||\mathbf{v}||_\infty^2$, $(b)$ follows from the fact that $\mathbf{H}$ does not change the Frobenius norm, and $(c)$ follows by substituting $\mathbf{A}$.

To bound the spectral norm, observe that

$$||\text{diag}(\mathbf{v})\mathbf{H}^T\mathbf{A}\mathbf{H}\text{diag}(\mathbf{v})||_2 \overset{(a)}{\leq} (||\mathbf{v}||_\infty)^2 ||\mathbf{H}^T\mathbf{A}\mathbf{H}||_2$$
$$\overset{(b)}{=} (||\mathbf{v}||_\infty)^2 ||\mathbf{A}||_2$$
$$\overset{(c)}{=} (||\mathbf{v}||_\infty)^2,$$

where $(a)$ follows by observing that each $\text{diag}(\mathbf{v})$ changes the spectral norm by at most $||\mathbf{v}||_\infty$, $(b)$ follows from the fact that rotation does not change the spectral norm, and $(c)$ follows by substituting $\mathbf{A}$. Since $\mathbf{v} = \mathbf{H}\mathbf{D}_3\mathbf{z}$, by McDiarmid's inequality, it can be shown that with probability $\geq 1 - 2de^{-d\epsilon^2/2}$, $||\mathbf{v}||_\infty \leq \epsilon||\mathbf{z}||_2$. Hence, by the Hanson-Wright inequality, we get

$$\Pr\left(\sqrt{d}\sum_{k=1}^d H(i,k)H(j,k)u^2(k) > t||\mathbf{z}||_2\right) \leq 2de^{-d\epsilon^2/2} + 2e^{-c\min(t^2/(d\epsilon^4),t/\epsilon^2)},$$

where $c$ is a constant. Choosing $\epsilon = (t/d)^{1/3}$ results in the theorem. $\qquad\square$

## Appendix E    Discrete Hadamard-Diagonal Structure in Binary Embedding

Motivated by the recent advances in using structured matrices in binary embedding, we show empirically that the same type of structured discrete orthogonal matrices (three blocks of Hadamard-Diagonal matrices) can also be applied to approximate angular distances for high-dimensional data. Let $\mathbf{W} \in \mathbb{R}^{D\times d}$ be a random matrix with i.i.d. normally distributed entries. The classic Locality

Sensitive Hashing (LSH) result shows that the sign nonlinear map $\phi : \phi(\mathbf{x}) = \frac{1}{\sqrt{D}} \operatorname{sign}(\mathbf{Wx})$ can be used to approximate the angle, i.e., for any $\mathbf{x}, \mathbf{y} \in \mathbb{R}^d$

$$\phi(\mathbf{x})^T \phi(\mathbf{y}) \approx \theta(\mathbf{x}, \mathbf{y})/\pi.$$

We compare random projection based Locality Sensitive Hashing (LSH) [4], Circulant Binary Embedding (CBE) [27] and Kronecker Binary Embedding (KBE) [29]. We closely follow the experimental settings of [29]. We choose to compare with [29] because it proposed to use another type of structured random orthogonal matrix (Kronecker product of orthogonal matrices). As shown in Figure 6, our result (HDHDHD) provides higher recall and lower angular MSE in comparison with other methods.

## References

[30] S. Chatterjee. Lecture notes on Stein's method and applications. 2007.