[Reviews · NeurIPS 2016]

Reviewer 1

Summary

The paper provides a new technique to construct Random Fourier Features (RFF), based on the polar decomposition of the linear transform defined by RFT.

Qualitative Assessment

I like very much this paper. It is well written and original. It is based on a simple idea, but very powerful. The experiments are designed to support the theory. To improve the content of the paper I suggest to add a small paragraph on Walsh-Hadamard matrices.

Confidence in this Review

2-Confident (read it all; understood it all reasonably well)


Reviewer 2

Summary

The paper proposes a way to improve approximations to a radial base kernel matrix by means of random projections used in the RFF method. Their method consists in using orthogonal projections. Empirical evidence support their claim. They also prove that the variance of a pointwise estimator decreases by the orthogonalization process. Finally, a faster approximation is presented that takes profit of the fast Hadamard transformation.

Qualitative Assessment

This is a very interesting, well written contribution with convincing results. Any idea how it compares to other approaches like Nystrom approximations, cf. ref[21]? Details: "Var" of \hat{K} was forgotten twice in LHS of (3) In proof of Lemma 3 line 284: the equality is not satisfied. Should it be an inequality? If so, how to proceed to line 285? Theorem 1, line 98: explain how to obtain "a uniform chosen random orthonormal basis" Theorem 1, line 100: "uniformly" should be skipped?

Confidence in this Review

2-Confident (read it all; understood it all reasonably well)


Reviewer 3

Summary

This paper tackles Gaussian kernel approximation (and even a broader range of random matrices) thanks to two novel techniques, that can be seen as upgrades of random Fourier features. The first method uses random orthogonal matrices while the second is based on structured discrete orthogonal matrices in order to speed up computation. The latter technique magically combines low storage requirements, low computation time thanks to structured matrices and low approximation error. The authors also give deep theoretical and experimental analyses.

Qualitative Assessment

Strength of the paper: The paper is very well written and interesting. It succeeds in explaining the breakthrough while giving many theoretical and experimental insights. The discovery is quite impressive since it breaks the no-free-lunch-theorem. Minor comments: - Line 72: this seems to be redundant with (and more obscure than) Line 121 - Line 78: why do the authors need D \le d for Lemma 1? - Line 94: please, explicitly write if f depends on some parameters (for instance d, see remark below). - Line 145: please, introduce a bit more the form HD_1HD_1HD_1. What are the previous references ? - Line 308: should not it be f(z) = \exp{z^2} z^4 (1 + (z^4+3z^2+1)/d)? Typos: - Line 80: on the matrix on the linear transformation - Line 87: basis - Line 108: \frac{Var(…)}{Var(…)} - Line 148: ORF'

Confidence in this Review

2-Confident (read it all; understood it all reasonably well)


Reviewer 4

Summary

This work deals with the Random Fourier Features (RFF) to explicit nonlinear feature maps with kernels, by proposing an orthogonal constructed basis.

Qualitative Assessment

A major issue in this work is that all the results are obtained for D=d, which is not the case in practice since D >> d. The authors propose to apply several times the linear transformation, as succinctly given in the end of Section 3. We found the explanation not enough, and we think that the influence of this trick should be thoroughly analyzed. Experiments given in Table 2 show that, while the conventional RFF outperforms the proposed techniques only on a single dataset for D=2d, the performance of the proposed techniques deteriorates when D increases. When D=8d and D=10d, the conventional RFF outperforms the proposed techniques on 3 out of the 6 datasets. Also on experiments, it is not clear why the conventional RFF outperforms the proposed techniques when dealing with the gisette dataset, namely for d = 4096. The authors should study the influence of d and the relevance of the proposed technique for d > 4096. In the lhs of eq. (3), there should be a ratio of the variances, not a ratio of the estimators. There are some typos, such as "forms a bases" -> forms a basis

Confidence in this Review

2-Confident (read it all; understood it all reasonably well)


Reviewer 5

Summary

This paper presents a very interesting finding, that using orthogonalized random projections significantly reduces the variance (error) of Gaussian kernel approximation. The author(s) further propose a structured variant to reduce the computational cost from O(d^2) to O(d log(d)) using the Walsh-Hadamard transformation. Some theoretical insights are presented to justify the procedure proposed. The gain over existing solutions is demonstrated with numerical experiments.

Qualitative Assessment

This is a well-written paper, the motivations and main results are clearly stated in the main text, and technical details are put in the supplements. I do however have a few questions needs to to clarified. 1). Line 98, what does it mean by 'uniformly chosen random orthonormal basis' ? I do not think the set of orthonormal basis is compact, how can there be an uniform measure defined on it? 2). Line 100, again, 'uniformly from N(0,I_d)' does not make sense. And why does w_i follow a Gaussian distribution? It does not seem to be trivial. Please clarify. 3). For very high dimensional data (large d), the random projections will be effectively orthogonal. Does the proposed scheme offer any advantage other than the computational efficiency in such a setting. 4). Line 308 (Supp), the extra term f(z) seems very large, but it is omitted in Eq. (3) (line 108). And in Figure 3, there is this 'theoretical curve' for the variance. I wonder how D is set in this Figure and how is this theoretical curve computed? And following might be some typos or negligences: Line 86 : I presume the 'orthogonal' actually means 'orthonormal'? Line 108 : Ratio of variance, shouldn't the 'K(x,y)' encapsulated by 'Var[]'? Line 202-206 : I guess poly(x) means polynomial, but I could not find them in the main text or supplements. ================================================================== I have read the author(s)' rebuttal which is also well-written, and my major concerns have been clarified. My score remains unchanged and I recommend the area chair to accept this paper.

Confidence in this Review

2-Confident (read it all; understood it all reasonably well)


Reviewer 6

Summary

The authors proposes ORF (orthogonal random features). The central idea is: by imposing orthogonality on the random feature matrix, we can significantly decrease the variance, and hence obtain a more accurate approximation of Gaussian RBF kernels. Theoretical analysis and empirical evaluations are provided.

Qualitative Assessment

This paper is technically well-written and easy to follow. The central idea of ORF is convincingly derived, explained, and experimented. Results of this paper also provide theoretical justification on similar observations in previous papers, e.g., Q. Le et al. (ICML’2013) observe that using Fourier Transform matrix gives better empirical performance. My only concern is on the generalization ability of the proposed method. The authors only consider Gaussian RBF kernels. The paper would be strengthened if the authors can show that the algorithm works under a more general scenario, e.g., for all shift-invariant kernels.

Confidence in this Review

2-Confident (read it all; understood it all reasonably well)